# Comprehensive Review on Synthesis, Properties, and Applications of Phosphorus (P^III^, P^IV^, P^V^) Substituted Acenes with More Than Two Fused Benzene Rings

**DOI:** 10.3390/molecules27196611

**Published:** 2022-10-05

**Authors:** Marek Koprowski, Krzysztof Owsianik, Łucja Knopik, Vivek Vivek, Adrian Romaniuk, Ewa Różycka-Sokołowska, Piotr Bałczewski

**Affiliations:** 1Division of Organic Chemistry, Center of Molecular and Macromolecular Studies, Polish Academy of Sciences, Sienkiewicza 112, 90-363 Łódź, Poland; 2Institute of Chemistry, Faculty of Science and Technology, Jan Długosz University in Częstochowa, Armii Krajowej 13/15, 42-200 Częstochowa, Poland

**Keywords:** acene, anthracene, phosphine, phosphonate, phosphonium salt, phosphorane, phosphate, diphosphene, tri-, tetra-, pentacoordinated phosphorus, properties

## Abstract

This comprehensive review, covering the years 1968–2022, is not only a retrospective investigation of a certain group of linearly fused aromatics, called acenes, but also a presentation of the current state of the knowledge on the synthesis, reactions, and applications of these compounds. Their characteristic feature is substitution of the aromatic system by one, two, or three organophosphorus groups, which determine their properties and applications. The (P^III^, P^IV^, P^V^) phosphorus atom in organophosphorus groups is linked to the acene directly by a P-C_sp2_ bond or indirectly through an oxygen atom by a P-O-C_sp2_ bond.

## 1. Introduction

Organophosphorus-substituted acenes are an increasingly important group of aromatic hydrocarbons due to the unique properties of the phosphorus atom, which can form tri-, tetra-, and pentacoordinated compounds. This creates the possibility of tuning the electronic properties of the aromatic system of acenes by substituting with organophosphorus groups with different electron characters, from electron-donating phosphine groups to strongly electron-accepting phosphonium groups. The acenes of this type, especially anthracenes, have usually been synthesized with the intention of applying them to organic light-emitting diodes (OLEDs) [1,2]. Other uses of these acenes include the synthesis of ligands for metal catalysts in the hydroformylation reaction [3] or in the Diels–Alder reaction as dienes [4]. Anthrylphosphonic acids and their derivatives have also been employed in the synthesis of self-assembled monolayers [5,6] and anthrylbisphosphonates were used to synthesize fluorescent organic nanoparticles as apoptosis inducers of cancer cells [7].

*Scope*: Acenes, as defined, are a group of aromatic compounds containing linearly fused benzene rings. In the literature, compounds with angularly fused benzene rings, such as phenanthrene, and compounds with fused heteroaromatic five- and six-membered rings are sometimes included in this group, and they are specifically called angularly fused acenes and heteroacenes, respectively. Both groups, due to the large number of combinations of ring types and the ring positioning in the fused acene, in addition to derivatives of the simplest linearly fused acene, naphthalene, are not the subject of this review. By virtue of their chemical structure, organophosphorus groups have not been restricted and include all groups containing three-, four-, and five-coordinate phosphorus atoms. In the compounds reviewed, organophosphorus groups contain a phosphorus atom, which is linked to the acene directly by a P-C_sp2_ bond or indirectly through an oxygen atom by a P-O-C_sp2_ bond, such as in phosphates. 

This review of the literature covers the period of 1968–2021 and reveals a lack of aromatics that contain more than three fused benzene rings (tetracenes, pentacenes, etc.). Therefore, the subject of this review is anthracenes substituted with any organophosphorus groups. 

In addition to the synthesis, this review discusses the reactions that these compounds undergo, and therefore, due to the difficulty of separating the two categories, “synthesis and reactions” are combined together in sections and subsections. Moreover, this review discusses the properties and applications of the synthesized derivatives as well. 

This manuscript is divided into five main sections covering phosphines, P^III^ acids derivatives, diphosphenes, phosphonates and phosphonic acids, phosphates, and the hetero analogs of the mentioned compounds. In the subsections, groups of compounds that can be obtained directly from the precursor included in the main section are identified, e.g., phosphine oxides, phosphonium salts, and phosphoranes can be obtained from phosphines by oxidation, alkylation, and halogenation, respectively, and therefore they are included in the section devoted to the synthesis and reactions of phosphines and derivatives. 

Finally, a comment on Section 3 is necessary. Phosphines (λ^3^-phosphanes), by definition, are P^III^ compounds containing a combination of three P-Z bonds (Z = C,H). Halophosphines AnthPX_2_ and Anthr_2_PX (X = F, Cl), which contain at least one P-C bond and one or two halogen atoms, are here classified as halides of the corresponding lower P^III^ acids [8]. Additionally included in this group are other representatives of the lower P^III^ acids, i.e., phosphonous acid diamides AnthP(NR_2_)_2_ (diaminophosphines) and phosphorous acid esters (phosphites). 

As a consequence of this division, phosphonous acid P^IV^ tautomers (*H*-phosphinic acids) and phosphinous acid P^IV^ tautomers (*H*-phosphine oxides) are also reviewed in this section. 

Diphosphenes, as the only compounds with a functional group containing more than one phosphorus atom, are discussed separately in Section 6. 

## 2. Synthesis and Reactions of Phosphines (AnthPR_2_) (Anth = Anthryl) and Derivatives 

This section discusses phosphines, which contain a three-coordinated phosphorus atom linked to three carbon atoms. The subsections of this chapter include groups of compounds that can easily be obtained from phosphine precursors by direct transformation to give compounds with tetra- and pentacoordinated phosphorus atoms. Phosphine oxides, sulfides, and selenides are placed together in one subsection because most papers simultaneously describe the synthesis and reactions of two or three groups of these derivatives. Secondary phosphine oxides (*H*-phosphine oxides) are discussed in Section 3.5 as phosphinous acid P^IV^ tautomers.

9-Bromoanthracene **1** is the most frequently used starting material for the syntheses of anthryl phosphines and other derivatives (cf. other subsections throughout this review).

A number of sterically shielded phosphorus ligands for metal catalysts were synthesized by Straub and co-workers via a selective stepwise nucleophilic substitution reaction at the phosphorus atom in triphenyl phosphite (Figure 1). Thus, first, diphenyl 9-anthrylphosphonite **2** was obtained in a 70% yield by substitution of one phenoxy group by anthryllithium, generated from 9-bromoanthracene **1** and *n*-butyllithium. Next, the second phenoxy group in triphenyl phosphite was replaced by 1-naphthyllithium to afford phenyl 9-anthryl(1-naphthyl)phosphinite **3** in an overall yield of 46%. Finally, the third least reactive phenoxy group was replaced by phenyllithium to give (anthryl)(naphthyl)(phenyl)phosphine **4** in a 66% isolated yield over three steps. In another synthetic sequence, [di(9-anthryl)](2-methoxyphenyl)phosphine **5** was obtained from triphenyl phosphite via phenyl phosphinite **6** by reacting triphenyl phosphite first with 9-anthryllitium, followed by 2-methoxyphenyllitium at −85 °C. Since phosphorus ligands, employed in homogeneous catalysis, often contain at least one sterically demanding substituent, this method delivers a strategy for the rapid and cost-efficient synthesis of such ligands [9]. 

Schmutzler et al. reported the synthesis of (anthryl)(diphenyl)phosphine **7** and trianthrylphosphine **8** starting from 9-bromoanthracene **1**, which was lithiated to give the intermediate 9-lithioanthracene **9**. The latter was reacted with 1 equiv. of chlorodiphenylphosphine or 1/3 equiv. of PCl_3_ in diethyl ether at reflux to give **7** or **8** in 71% and 7% yields, respectively. The irradiation of **7** in the presence of W(CO)_6_ with a mercury lamp for 34 h formed the pentacarbonyltungsten complex **10** in an 18% yield (Figure 2) [10].

Chan and co-workers reported a simple monophosphinylation reaction of 1,8-dichloro-anthracene **11**, leading to 8-chloro-1-(diphenylphosphino)anthracenes **12** in a 33% yield (Figure 3) [11]. This reaction was carried out in DMF at 160 °C for 60 h and catalyzed by 10 mol% of palladium supported on charcoal in the presence of 5 equiv. of triphenylphosphine. The addition of 5 equiv. of sodium iodide improved both the rate and yield of the reaction. The second chlorine atom in **11** was not substituted in this reaction and only the mono-derivative **12** was observed, despite the presence of an excess of triphenylphosphine. The authors claimed that steric hindrance of the diphenylphosphino group in position 1 of anthracene protected against the reaction of the second chlorine atom. 

Misochko and co-workers [12] obtained 9-(1-phosphirano)anthracene **13** from 9-phosphinoanthracene **14** and ethylene glycol ditosylate as substrates by adapting the procedure of Robinson et al. [13] (Figure 4). In the next step, 9-(1-phosphirano)anthracene **13** was subjected to UV photolysis to receive a stable triplet anthrylphosphinidene **15**, which could be characterized by electron paramagnetic resonance (EPR). 

Che and co-authors reported the rhodium(I)-catalyzed C−H arylation of 9-(diphenylphosphino)anthracene **7**, as an example of functionalization of phosphines, to give 1-aryl-substituted derivatives **16a** and **16b** (Figure 5) [14]. The presented strategy provided access to *peri*-substituted (naphth-1-yl)phosphines as well. 

The synthesis of optically active 1,2-ethylene bis(phosphine) (*S*,*S*)-**17**, presented by Maienza and co-workers, is the only example of a molecule containing two anthrylphosphino moieties linked via an alkyl linker [15]. First, 9-(dichlorophosphino)anthracene **18** was utilized in the reaction with methyl magnesium bromide and BH_3_·SMe_2_ to obtain 9-anthryldimethylphosphine borane **19**. Then, **19** was enantioselectively deprotonated in the presence of (*−*)-sparteine with *s*-BuLi and then oxidatively coupled with Cu (II) to give a mixture of enantiomers of 1,2-ethylene bis(phosphine) diboranes (*S*,*S*)-**20** and (*R*,*S*)-**20** in a 6:1 ratio and 70% total yield (Figure 6). Diastereomerically pure (*S*,*S*)-**20** was received by crystallization from a CH_2_Cl_2_/Et_2_O mixture in a 39% yield and 18% ee. Finally, the diphosphine borane (*S*,*S*)-**20** was deprotected by stirring in morpholine at room temperature for 12 h. In this reaction, the 1,2-ethylene bis(phosphine) (*S*,*S*)-**17** was obtained and its enantiomeric excess was determined based on the corresponding phosphine oxide, which was prepared by oxidation of **17** with an excess of H_2_O_2_. 

In the reviewed literature, the synthesis, transformations, and utilization of anthracenes with two phosphino groups on the aromatic moiety were found and are presented below. 

1,8-Bis(diphenylphosphino)anthracene **21** was synthesized in a three-step reaction in a 51% overall yield starting from 1,8-dichloro-9,10-anthraquinone **22** by Haenel and co-workers (Figure 7). The anthraquinone **22** was converted to 1,8-difluoroanthracene **23** by chlorine-fluorine exchange to give **22a** followed by reduction with zinc, from which **21** was obtained by reacting it with potassium diphenylphosphide [16]. 

Gelman and co-workers presented the quantitative Diels–Alder cycloaddition of 1,8-bis-(diphenylphosphino)anthracene **21** to diethyl fumarate **24a**. The adduct **25a** was used for the synthesis of bifunctional PCsp^3^P pincer catalyst for the acceptorless dehydrogenation (CAD) of the primary and secondary alcohols to give carbonylic and carboxylic compounds (Figure 8) [17]. 

The same research group reported a synthetic scheme that relied on the carbo-Diels–Alder reaction cycloaddition of 1,8-bis-(diphenylphosphino)anthracene **21** to enantiomerically pure bis-(methyl-(*S*)-lactyl) fumarate **24b**, leading to the formation of chromatographically resolvable diastereomers **25b** that could be converted into a pair of enantiomerically pure antipodes (Figure 8) [18]. 

Jiang and co-workers [19] showed a practical utilization of 9,10-bis(diphenylphosphino)anthracene **26**. They obtained a red-light-controllable soft actuator, which was driven by the low-power excited triplet−triplet annihilation-based upconversion luminescence. This system consisted of 9,10-bis(diphenylphosphino)anthracene **26** and the Pt(II) tetraphenyltetrabenzoporphyrin complex **27** (Figure 9). It was then incorporated into a rubbery polyurethane film and assembled with an azotolane-containing film to study its possible utilization as a highly effective phototrigger of photodeformable cross-linked liquid-crystal polymers. In this system, the Pt(II) complex **27** acted as a sensitizer, whereas **26** was an annihilator, which induced *trans-cis* photoisomerization of azotolane **28** and **29**. The authors achieved a highly effective red-to-blue triplet-triplet annihilation-based upconversion layout with a low-energy excitation light source, large anti-Stokes shift (165 nm), and high absolute quantum yield (9.3%). 

In the literature reviewed, the chemistry of anthracenes with two or three phosphino groups and their mixed tetracoordinated derivatives was also found and is presented below. 

Thus, 9-bromo-1,8-bis(diisopropylphosphino)anthracene **30**, repulsively interacting with 1,8,9-tris(phosphino)anthracene **31** (Figure 10), single donor stabilized 8-diisopropylphosphino-1-thiophospinoyl-9-metathio/metaselenophosphono)anthracenes **32** and **33** (Figure 11), and doubly phosphine donor stabilized phosphenium salt **34** (Figure 12), were synthesized by Kilian and co-workers [20]. The attempted introduction of the third phosphorus atom at the position 9 via Br/Li exchange followed by the reaction with chloro-bis(dimethylamino)phosphine resulted in formation of 1,8,9-tris- and 1,8-bis(phosphino)anthracenes **31** and **35**, respectively. The derivative **31** had two relatively inert and bulky diisopropylphosphino groups at positions 1 and 8 whilst the third reactive phosphino group with two P–N bonds on the middle ring opened up the possibility of various transformations on this phosphorus atom, situated in a very crowded surrounding. 

Alcoholysis of **31**, leading to the intermediate **36**, followed by oxidation with sulfur or selenium, afforded phosphine donor stabilized anthracenes **32** or **33** with metaphosphono groups, respectively. 

Further reaction of **31** with diphosphorus tetraiodide in 1,2-dichloromethane gave the chlorophosphenium cation **34** stabilized by two phosphino donors at positions 1 and 8, forming a linear P–P–P arrangement. In the first step, the phosphino-phosphonium cation **37** was formed as a transient species. In the next step, the dimethylaminophosphino group was substituted by the chloride anion, which was available from the I/Cl halogen exchange reaction in chlorinated solvent (DCM). Iodide and iodine (I_2_) and triiodide originated from disproportionation reaction of P_2_I_4_. 

### 2.1. Phosphine Oxides (AnthP(=O)R_2_), Phosphine Sulfides (AnthP(=S)R_2_), Phosphine Selenides (AnthP(=Se)R_2_)

Phosphine oxides, sulfides, and selenides are placed together in one subsection because most papers simultaneously describe the synthesis and reactions of two or three groups of these derivatives. All of them were obtained directly from the corresponding phosphines. 

A new method for the synthesis of phosphine oxides was published by Yang et al. [21]. They employed a cross-coupling reaction and showed that aryl, vinyl, and benzyl-ammonium triflates reacted with the corresponding phosphorus-based nucleophiles in the presence of the nickel catalyst NiCl_2_/dppf, (dppf = 1,1’-bis(diphenylphosphino)ferrocene) (Figure 13). 

The counterion played a minor role in this reaction, so it could be replaced with chloride, bromide, iodide, mesylate, or tosylate anions without a significant loss of yield. A considerable advantage of this synthesis was that ammonium salts are cheap and readily viable. 

Another method, which was developed by Zhang et al. [22], involved a direct transformation of aromatic acids into the corresponding phosphine oxides in the presence of palladium(II) salts. Several substrates were shown to react in this manner, including 2-anthroic acid **40**, which was transformed to 2-(diphenylphosphinoyl)anthracene **39** (Figure 14), providing an alternative to the synthesis of the latter from anthryl ammonium triflate **38** (Figure 13), although in significantly lower yields. 

The optimal temperature for the reaction was 115 °C. Changing the temperature reduced the yield, as did changing the solvent to a highly polar one (DMF, *N*,*N*-dimethylformamide). 

Another study carried out by Zhao and coworkers [1] showed that 9,10-bis(diphenylphosphinoyl)anthracene **41** could readily be synthesized from 9,10-dibromoanthracene **42** by substitution of chlorine in chlorodiphenylphosphine with 9,10-dilithioanthracene obtained from a double Br/Li exchange in **42** followed by oxidation of the resulting bis(diphenylphosphino)anthracene **26** with hydrogen peroxide to yield **41** in a 47% yield. 9,10-Dibromoanthracene **42** was obtained by bromination of anthracene in chloroform (Figure 15). 

The product **41** has been synthesized with the intention of applying it in organic light-emitting diodes (OLEDs). Zhao et al. found that **41** was a yellow-green solid with a melting point that reached 265 °C. A similar study by Tao and co-workers [23] confirmed the fluorescent properties of **41**, which could be used in the construction of OLEDs. The authors also claimed that the conversion from **26** to **41** was the first reported triplet-triplet annihilation system activated by hydrogen peroxide. 

Furthermore, Xu and co-workers [24] linked the increase in fluorescence properties within the **26**/H_2_O_2_ system to the degree of photooxidation. This system could be used as an indicator of the reaction time, oxygen exposure, and light irradiation or a time-oxygen and light indicator (TOLI). The compound **41** was also the subject of interest in another study by Xu et. al. [25], who used it to investigate the properties of a samarium complex Sm(hfac)_3_(**41**)_3_ (hfac = hexafluoroacetylacetonato). 

Wu and co-workers [2] showed the synthesis and properties of (9,10-diphenyl-2-phosphinoyl)anthracene **43** with the intention of using it as a true-blue OLED material. The synthesis was similar to the one proposed by Zhao and coworkers [1]: it included treatment of 2-bromo-9,10-diphenylanthracene **44** with *n*-BuLi, followed by the addition of chlorodiphenylphosphine and subsequent oxidation (Figure 16). 

The compound **43**, which readily crystallized as a light-yellow solid, was obtained in a 30% yield. It exhibited red-shift fluorescence compared to 9,10-diphenylanthracene. This study showed that **43** was not suitable for the true-blue OLED material due to the fact that the compound exhibited a red shift. 

Yamaguchi et al. [26] reported a synthesis and photochemical characterization of tri(9-anthryl)phosphine **8** and tri(9-anthryl)phosphine oxide **45** (Figure 17). 

Tri(9-anthryl)phosphine **8** was synthesized from 9-bromoanthracene **1** by treatment with *n*-butyllithium, followed by the addition of phosphorus trichloride. Tri(9-anthryl)phosphine oxide **45** was obtained by oxidation of **8** with hydrogen peroxide (Figure 17). The tri-coordinated **8** and tetra-coordinated derivative **45** exhibited a weak fluorescence. 

The synthesis of phosphine oxides **46a**–**h** containing carboxylic acid esters and 9,10-dihydro-9,10-ethanoanthracene moiety, described by Okada and co-workers, is an example of utilization in the synthesis of 2-(dipenylphospinoyl)anthracene **39**. The bulky compounds **46a**–**h** were synthesized in the reaction of **39** with dimethyl methylene malonate, dimethyl fumarate, or methyl acrylate and paraformaldehyde in 21–95% yields. (Figure 18) [27]. 

Katagiri et al. [28] synthesized 9-(diphenylphosphinoyl)anthracene **47** and 9-(anthrylphenylphosphinoyl)anthracene **48** from diphenyl chlorophosphine and phenyl dichlorophospine, respectively (Figure 19). The synthesis was analogous to the previous method by Schwab and co-workers [29] and first required halogen/lithium exchange and then oxidation with H_2_O_2_. The authors revealed that phosphine oxides **47** and **48** did not lead to the formation of a photodimer in the solid state, whereas in chloroform or acetonitrile under an N_2_ atmosphere, at the 365-nm-wavelength irradiation, the [4π + 4π] photodimerization of **47** occurred to give **49**. In addition, the absorption and emission spectra of the compound **47** in acetonitrile showed characteristic absorption and emission bands of anthryl moieties while photodimerization of the anthryl groups led to the disappearance of these bands. The authors reversibly returned **49** to **47** by heating the probe at 80 °C. In contrast, analogous irradiation under an O_2_ atmosphere resulted in the formation of anthraquinone **50**. 

During attempts to obtain a “masked” version **51** of the phosphaalkene Mes-P=CH_2_, Gates and co-workers [30] synthesized 9-[(methyl)(mesityl)phosphino)]anthracene **52** and the corresponding 9-[(methyl)(mesityl)phosphinoyl)]anthracene **53** in the reaction between (chloro) (chlorometyl)(mesityl)phosphine **54** and the anthracene magnesium (MgAnth•3THF) in THF. The expected adduct **51**, which was initially formed, then decomposed to give the anthracene derivative **52** in an 83% yield. The latter was oxidized to **53** in a 9% yield only (Figure 20). 

Wang and Zhu reported the palladium-catalyzed decarbonylation of 9-[(1-keto diphenylphosphinoyl)]anthracene **55** in the presence of 1 mol% of Pd_2_(dba)_3_ and 8 mol% of the phosphine ligand (PCy_3_) to give 9-(diphenylphosphinoyl)anthracene **47** in an 80% yield [31] (Figure 21). 

Drabowicz and co-workers synthesized optically active 9-[(*t*-butyl)(phenyl)phosphinoyl)]anthracene **56** in a 71% yield in the Hirao reaction of palladium-catalyzed cross-coupling reaction of 9-bromoanthracene **1** with optically active *t*-butylphenylphosphine oxide **57** (Figure 22) [32]. The formation of carbon–phosphorus bonds took place with retention of the configuration, and the stereoretention of this reaction was confirmed by X-ray analysis. 

Stalke and co-workers synthesized three positional isomers of 1-, 2-, and 9- (diphenylthiophosphinoyl)anthracenes **58**, **59**, and **60** that revealed a solid-state fluorescence in three different colors with differences in emission wavelengths of over 100 nm. Analysis of the solid-state structure of **59** and photophysical properties allowed the unusual yellow emission to be attributed to the formation of excimer in the solid state. Therefore, substitution at position 1 of the anthracene fluorophore with suitable substituents may be a promising strategy to obtain long wavelength emission in the solid state using structurally easy to modify compounds (Figure 1) [33]. 

Schillmöller and co-workers [34] synthesized four 9-(diphenylthiophosphinoyl)anthracenes **58** and **61**–**63** with alkyl and phenyl substituents at the position 10 via sulfurization of 9-(diphenylphosphino)anthracenes **7** and **64**–**66**. The latter were obtained from the corresponding bromoanthracenes **1** and **67**–**69** (Figure 23) [35,36]. 

The compounds **58** and **61**–**63** were crystallized and their X-ray structures were then determined. These studies revealed that oxidation of the phosphorus atom with sulfur significantly changed the molecular structural parameters and the crystal packing. This caused a strong bathochromic shift, which resulted in a green solid-state fluorescence. Moreover, the authors prepared four host-guest complexes, with **62** as a host molecule and benzene, pyridine, toluene, and quinoline as guest molecules. This resulted in enhanced emission and up to a five times higher quantum yield in comparison to the pure compound **62**. 

Walensky et al. characterized 9-(diphenylthio- and diphenylselenophosphinoyl)anthracenes **58** and **70**, respectively, by NMR and optical spectroscopy (Figure 24) [37]. The authors demonstrated that ^31^P NMR shifts for **58** and **70** were shifted upfield when compared to the unoxidized analog. This was due to the loss of planarity and relatively greater σ- than π-bonding between the phosphorus atom and the anthracene carbon. 

When excited at 310 nm, compounds **58** and **70** showed emission similar to that of unsubstituted anthracene, displaying peaks at 380, 402, 430, and 450 nm. When the excitation wavelength was shifted to 410 nm, the observed emission became structureless and was red-shifted by around 50 nm relative to the typical emission of the unsubstituted anthracene. The authors did not observe excimer formation for these compounds. Moreover, a small deviation from planarity in the anthracene ring was observed for **58** and **70** and the angle of deflection was 3° and 5°, respectively. 

Schwab and co-workers [38] obtained 9-bromo-10-(diphenylphosphino)anthracene **71** and its thio- **72** and seleno- **73** derivatives (Figure 25). In the first stage, 9,10-dibromoanthracene **42** was treated with *n*-BuLi followed by the addition of chlorodiphenylphosphine to give 9-bromo-10-(diphenylphosphino)anthracene **74**. Then, **74** was oxidized to the corresponding oxo-, thio-, and seleno derivatives **71**–**73** in high yields according to the procedure described by Stalke et al. [39]. Their spectral and structural properties were investigated and shown to be largely consistent with those of the 9,10-diphosphino derivatives **80**–**82** mentioned below (Scheme 27). 

In an analogous manner, the same authors [29] synthesized bulky 9-(diisopropylphosphino)anthracene **75**, 9-bromo-10-(diisopropylphosphino)anthracene **76**, and 9,10-symmetrically-substituted anthracene **77**, which were then oxidized to the corresponding derivatives **78a**–**f** and **79** (Figure 26). 

9,10-Bis(diphenylphosphinoyl)anthracene **80**, 9,10-bis(diphenylthiophosphinoyl)anthracene **81**, and 9,10-bis(diphenylselenophosphinoyl)anthracene **82** were obtained by oxidation (E = O, S, Se) of 9,10-bis(diphenylphosphino)anthracene **26** again using H_2_O_2_•(NH_2_)_2_C=O (urea) (dichloromethane, 0 °C), elemental sulfur (toluene, reflux), and selenium (toluene, reflux), respectively (Figure 27). The compounds obtained were significantly more soluble in organic solvents than the starting material **26**. The absorption and emission spectra of **80**–**82** were recorded in solution and in the solid state. In solution, only **80** exhibited a detectable emission whereas **81** did not emit. The latter showed strong fluorescence in the solid state at λ = 508 nm. This molecule formed single crystals possessing a groove suitable for binding toluene reversibly to the anthracene chromophore by means of C-H⋯π-ring center interactions. Hence, the crystalline **81** was the first solid-state excimer that could serve as a chemosensor to detect toluene selectively. Single crystals of **80** emitted at λ = 482 nm, whereas the selenium derivative **82** did not emit in the solid state. In addition, the crystal structures of compounds **80**–**82** were analyzed using the single-crystal X-ray diffraction technique (Figure 27) [39]. The substrate **26** was obtained based on the procedure described by Prabhavathy and co-workers [40]. 

### 2.2. Phosphine Boranes (AnthPR_2_•BH_3_) 

In this subsection, the presented syntheses and reactions of phosphine molecules with P-B coordinate (semipolar, dative) bonds are presented. 

The ring opening of enantiomerically pure oxazaphospholidineborane **83** with bulky anthryllithium to give phosphineboranes **84** was studied by Stephan and co-workers. The authors proposed an explanation for the low 4% yield of **84**. They reported that in the case when the attack on the phosphorus atom was hindered, deprotonation of the benzylic proton occurred, yielding *trans*-(*N*-methylamino)(phenyl)(1-phenyl-1-propenyloxy)phosphineborane **85** instead of **84** (**84**/**85** = 5/95) (Figure 28) [41]. 

1,8-Bis-(diisopropylphosphino)-9-methoxyanthracene **86**, as a starting material for the synthesis of 9-boron-substituted 1,8-bis-(diisopropylphosphino)anthracene **87a**/**87b**, was prepared by Akiba and co-workers by treatment of 1,8-dibromo-9-methoxy-anthracene **88**, first with *n*-BuLi and then with diisopropylchlorophosphine. Diphenylchlorophosphine was used as well; however, only the diisopropylphosphine derivative **86** could be successfully transformed into 1,8-bis(diisopropylphosphino)-9-bromoanthracene **30** with LDBB (lithium di-*tert*-butylbiphenylide) followed by treatment with BrCF_2_CF_2_Br, as a brominating agent, in a 51% yield. The introduction of a boron substituent at the position 9 in **30** via Br/Li exchange followed by reaction with chloroborane **89** led to the formation of 1,8-bis(diisopropylphosphino)-9-borylanthracene **87a**/**87b**. The ^1^H and ^31^P NMR spectra of **86** showed a symmetrical anthracene pattern at room temperature. This meant that a very rapid bond switching process between **87a** and **87b** occurred in solution (Figure 29) [42]. 

### 2.3. Phosphine–Metal Complexes (AnthPR_2_-Metal) 

In this subsection, complexes of phosphines possessing at least one anthryl substituent with metals, such as Au, Ag, Au/Ag/Sb, Fe, Pd, Pt, Ir, Lu, Eu, Ru, and Ni, are reviewed. 

Other metals (W, Os, Co), i.e., the pentacarbonyltungsten complex of 9-diphenylphospino)anthracene, are reported in Section 2 while the triosmiumdodecacarbonyl cluster and dinuclear cobalt complex are discussed in Section 3, respectively. 

Gold and platinum(II) complexes of the phosphine ligands PAnth_n_Ph_3-n_ (Anth = anthryl) were synthesized by Mingos et al. [43]. The authors recorded ^31^P{^1^H} NMR chemical shifts for (anthryl)(diphenyl)phosphine, (dianthry)(phenyl)phosphine and trianthrylphosphine, their oxo derivatives, and gold (I) halide and gold (I) nitrate complexes. Moreover, a crystal structure of the [AuCl(PAnth_2_Ph)]•CH_3_Cl complex was determined by X-ray analysis. An example of the preparation of the gold (I) complex [Au(NO_3_)(PAnthPh_2_)] **90** obtained from 9-(diphenylphosphino)anthracene **7** is shown in Figure 30. 

Several other Au and Pt complexes were also synthesized: *trans*-[PtCl_2_(PAnthPh_2_)_2_] (64%); *trans*-[Pt (CH_3_CN)_2_(PAnthPh_2_)_2_](BF_4_)_3_ (74%); *trans*-[Pt (CH_3_CN)_2_(PAnth_2_Ph)_2_](BF_4_)_3_ (68%); [Au(NO_3_)(PAnthPh_2_)] (95%); [Au(NO_3_)(PAnth_2_Ph)] (79%); [Au(NO_3_)(PAnth_3_)] (53%); [AuCl(PAnthPh_2_)] (91%); [AuCl(PAnth_2_Ph)] (93%); and [AuCl(PAnth_3_)] (72%). 

A luminescent molecular metalla(Au)cyclophane **91**, which was synthesized from the self-assembly of the molecular “clip” **92** and bipyridine, showed a large rectangular cavity of 7.921(3) × 16.76(3) Å (Figure 31). The electronic absorption/emission spectroscopy and electrochemistry of **91** were studied. The **2**^4+^ ions were self-assembled into a 2D mosaic in the solid state via complementary edge-to-face interactions between phenyl groups. ^1^H NMR titrations ratified the 1:1 complexation of the cations **91** and various aromatic molecules. Comparison of the structures of the inclusion complexes indicated an induced-fit mechanism operating in the binding. The luminescence emission of **91**^4+^ could be quenched upon the guest binding. The binding constants were determined by both ^1^H NMR and fluorescence titrations. Solvophobic and ion-dipole effects were shown to be important in stabilizing the inclusion complexes [44]. 

Complexes **92** (X = OTf^−^, ClO_4_^−^, PF_6_^−^, BF_4_^−^), as discrete binuclear, trinuclear, and tetranuclear metallacycles, were isolated and characterized, showing novel puckered-ring and saddles-like structures in the tri- and tetranuclear metallacycles (Figure 31) [45]. 

The trinuclear Au(I) complex [Au_3_(PAnthP)_3_][ClO_4_]_3_ **93** was synthesized by Yip and co-workers in the reaction of 9,10-bis-(diphenylphino)anthracene (**P**Anth**P**) **26** and 1 equiv. of Me_2_S AuCl in methanol at reflux. The authors observed a stable gold ring in the solution and no NMR signals arising from the free ligand (Figure 32) [40]. 

The UV/Vis absorption spectra of **26** and its complex **93** showed intense bands at 396 and 424 nm assigned to ^1^π-π* transitions in the anthryl ring. Excitation of CH_3_CN solution of **93** at 400 nm gave an emission at 475 nm with a quantum yield of Θ = 0.05. 

The reaction of **26** (**P**Anth**P**) with 2 equiv. of Me_2_SAuX in CH_2_Cl_2_ led to the new binuclear complexes (*µ*-**P**Anth**P**)(AuCl)_2_ **94a** and (*µ*-**P**Anth**P**)(AuBr)_2_ **94b** with Au(I)−X−Ag(I) halonium cation (Figure 33) [46]. The reaction of **94a** and **94b** with 2 equiv. of AgSbF_6_ led to spontaneous formation of the [(*µ*-**P**Anth**P**)-Au_2_]^2+^ ion, and then, after the addition of AgSbF_6_ (0.5 equiv.) in a THF solution, gave crystals of {[(*µ*-**P**Anth**P**)(AuCl)_2_]2Ag}^+^SbF6^−^ **95a** and {[(*µ*-PAnthP)(AuCl)_2_]2Ag}^+^SbF_6_^−^ **95b** as products.

9,10-Bis(diphenylphosphino)anthracene (PAnthP) **26** with two donor phosphorus atoms has been used as a P-ligand unit for metals. Thus, the double-helicate dinuclear silver(I) complex, [Ag_2_(4’-Ph-therpy)_2_](SO_3_CF_3_)_2_, was reacted with **26** to give the corresponding dinuclear complex **96** (4’-Ph-therpy = 4’-phenyl-terpyridine). The latter showed a strong fluorescence in the solid state with an excitation band at 383.5 nm, emission band at 535.5 nm, and lifetime of 4.20 ns, but the derived complexes did not show fluorescent properties (Figure 2) [47]. 

An iron(0)tetracarbonyl complex **97** was synthesized from di(9-anthryl)fluorophosphine **98**, which was stable to redox disproportionation (Figure 34) [48]. 

Pincer iridium complexes **99** derived from 1,8-bis(diphenylphosphino)anthracene **35** turned out to be suitable platforms for the C−H activation of methyl *tert*-butyl ether (MTBE) (Figure 3) [49]. 

A series of thermally stable Ir, Ni, and Pd complexes were obtained from 1,8-bis(dialkyl and diphenylphosphino)anthracenes **100**, **35**, and **21**. The anthracenes **100** and **35** were prepared similarly to **21** by direct nucleophilic substitution of fluorine atoms in 1,8-difluoroanthracene by potassium di-*tert*-butylphosphide or potassium di-*iso*-propylphosphide. The reaction of **100** with IrCl_3_•3H_2_O in 2-propanol/water afforded the complex **101** as a red crystalline powder in an 86% yield (Figure 35). The reduction of **101** under a hydrogen atmosphere gave mixtures of the yellow-colored iridium tetrahydride **102** and the red-colored iridium dihydride **103**. By saturating solutions of such mixtures with hydrogen, the equilibrium was shifted towards **102**. Evaporation of the solvent under vacuum resulted in the formation of the analytically pure complex **103** in a >95% yield. The thermally stable complexes **103** and **104** were ideal for homogeneous catalysts in the alkane dehydrogenation above 200 °C. The complex **103** in alkane solution was stable at 250 °C and catalyzed the dehydrogenation reactions at this temperature [50]. 

Osawa et al. [51] synthesized bis[(9-diisopropylophosphino)anthracene]-tris(hexafluoroacetylacetonato)europium(III) **108** (Figure 36). First, the authors prepared 9-(diisopropylphosphino)anthracene **78a** (Figure 26) according to the Schwab et al. protocol [29], which was next reacted with tris(hexafluoroacetylacetonato)europium(III) **108a** for 8 h in refluxing methanol solution to obtain **108** in a 55% yield after recrystallization. 

Osawa and co-workers determined the crystal structure of the Eu(III) complex **108** and studied its intra-complex energy transfer. The studies revealed that laser irradiation of this compound in *n*-hexane gave blue emission, which was ascribed only to the 9-(diisopropylphospino)anthracene moiety, not to the central Eu(III) ion (Figure 36). 

Kitagawa and co-workers [52] obtained a novel coordination polymer **109** based on 9,10-diphenyl-2,6-bis(diphenylphosphinoyl)anthracene **110** as a core and two molecules of Lu(hexafuoroacetylacetonate)_3_ that interacted with the core (Figure 37). 

First, the anthracene **110** was synthesized from 2,6-dibromo-9,10-diphenylanthracene **111** and diphenylphosphine. The first step of the synthesis was performed in the presence of potassium acetate and palladium acetate, and next the resulting bisphosphine intermediate was oxidized to **110** in a 28% yield. The polymer **109** was prepared in a microtube by the liquid-liquid diffusion-assisted crystallization method. The authors studied the photophysical properties and thermal stability of **109** and its oxide **110**. The luminescence quantum yield was enhanced from 18% up to 25% (λ_ex_ = 380 nm) due to the introduction of Lu(hexafuoroacetylacetonate)_3_ molecules into the phosphine oxide system, as a result of which bright, pure sky-blue emission was observed. In addition, the compound **109** showed a higher temperature of decomposition (340 °C) than **110**. 

The diphosphine-bridged dimer of the oxo-centered triruthenium–acetate cluster unit [{Ru_3_O(OAc)_6_(py)_2_}_2_(dppan)](PF_6_) **112** was synthesized by Chen and his co-workers (Figure 38). The reaction of [Ru_3_O(OAc)_6_(py)_2_(CH_3_OH)](PF_6_) with 9,10-bis(diphenylphosphino)anthracene (dppan) **26** resulted in the formation of **112** in a 67% yield. The redox studies of the complex **112** revealed the presence of electronic communication between two triruthenium units mediated through bridging dppan [53]. 

A number of tri-, tetra-, and penta-ruthenium clusters **113**–**115** were synthesized by Deeming and co-workers. When a suspension of [Ru_13_(CO)_12_] and a slight excess of 9-(diphenylphosphino)anthracene **7** in octane were heated to reflux at 125 °C for 4 h, several products were obtained, including the yellow trinuclear cluster [Ru_3_(µ-H)_2_(CO)_8_(µ_3_- C_14_H_7_PPh_2_)] **113** and the purple tetraruthenium butterfly complex [Ru_4_(CO)_11_(µ_4_-C_14_H_7_PPh^2^)] **114**. Both anthracyne complexes and also the dark purple pentaruthenium bow-tie cluster, [Ru_5_(CO)_13_(µ_5_-η^1^:η^2^:η^3^: η^3^-C_14_H_8_- η^1^- PPh_2_)] **115** were obtained via the double metallation from one of the unsubstituted rings (Figure 39). Furthermore, treatment of the trinuclear species **113** with 1 equivalent of [Ru_3_(CO)_12_] in refluxing octane resulted in a cluster build-up, with the formation of the tetra- and penta-ruthenium species **114** and **115**. Likewise, the thermolysis reaction of **114** with [Ru_3_(CO)_12_] also led to **115**. The crystal structure of **3** revealed a unique µ_5_-interaction of the ligand with the ruthenium cluster [54]. 

The bulky phosphine ligand **116** was prepared by Claverie et al. and used to generate the phosphine palladium complex **117**. The complex catalyzed ethene polymerization to yield linear polyethene; however, its catalytic activity was smaller compared to complexes with phenyl, naphthyl or phenanthryl substituents, which corresponded to increasing cone angles and decreasing basicity (Figure 40) [55]. 

Yamamoto and Shimizu synthesized 9-(diphenylphosphino)anthracene-based palladacycles **118a** and **118b** that catalyzed conjugate addition of arylboronic acids to electron-deficient alkenes, such as α,β-unsaturated ketones, esters, nitriles, and nitroalkenes. The monomeric catalysts, which were synthesized from K_2_PdCl_4_, 9-(diphenylphosphino)anthracene, and trialkyl phosphites, exhibited turnover numbers of up to 700 (Figure 4) [56]. 

Mingos and co-workers reported the synthesis and structural characterization of the Pd complex [Pd(dba)L_2_] **119** (where L = **120** and dba = dibenzylideneacetone) obtained from [Pd_2_(dba)_3_] and the corresponding 1-(diphenylphosphino)anthracene **120** (L) (Figure 41). The single-crystal X-ray structural analyses confirmed that these complexes adopted a trigonal planar structure, with the dba ligand coordinated by a double bond [57]. 

Dibenzobarrelene-based C(sp^3^)-metallated pincer complexes **121a**, **121b**, and **121c** were synthesized by the Diels–Alder [4 + 2] cycloaddition reaction of organometallic anthracene dienes **122a**, **122b**, and **122c** with dimethyl alkyne dicarboxylate as a dienophile (Figure 42). This straightforward approach has an advantage over traditional synthetic routes, such as either C–H activation or oxidative insertion of a coordinated transition metal into the C–X bond of the halogenated spacer [58]. 

A number of metal complexes **122a**, **122b**, and **123** have been synthesized using 1,8-bis(diphenylphosphino)anthracene **21** as a ligand (Figure 43). The latter was synthesized from dipotassium 1,8-anthracenedisulfonate **124** and potassium diphenylphosphide (Ph_2_PK). The reaction of **21** with nickel(II) chloride or bis-(benzonitrile)palladium(II) chloride led to cyclometallation of the anthracene C-H bond at 9-position and resulted in the formation of square-planar chelate complexes **122b** or **122a**, respectively. Treating the complex **122b** with aqueous potassium cyanide did not remove nickel from **122b** but converted **122b** into **123** by substituting chloride with cyanide, confirming the high stability of these cyclometallated chelate complexes. The strong metal bonding in **122b** made it an ideal ligand for the development of new catalysts. Like the anthracene unit in **21**, other polycyclic acenes or heteroarenes might also be useful as a rigid backbone for bidentate phosphines [59]. 

The platinum (II) complex **125** and photochemically dimerized product **126** were synthesized from 9-(difluorophosphino)anthracene **127** (Figure 44), obtained in the reaction of anthryllithium with chlorodifluorophosphine, with the former being synthesized in the reaction of *n*-butyllithium with 9-bromoanthracene **1**. The dimer **126** constituted one of the six possible rotational isomers. A rotation of the PF_2_ group was hindered by strong F-H interactions at temperatures up to at least 105 °C [60]. 

Hu et al. [61] described the cycloplatination reaction of 9,10-bis(diphenylphosphino)anthracene **26** with Pt(bis(diphenylphosphino)methane)(OTf)_2_ **128** to give [Pt(bis(diphenylphosphino)methane)(9-(diphenylphosphino)anthracene)PO-H)]OTf **129**. The uncoordinated P atom in the complex was oxidized when exposed to air (Figure 45). 

The same authors also studied the influence of the reaction conditions on the regioselectivity of the double cyclometallation process (Figure 46) [62]. 

Other dicyclometalated complexes *syn*- and *anti*-[Pt_2_(L)_2_(PAnthP-H_2_)](OTf)_2_(Pt_2_) (Anth = anthrylene) **130**–**132** have been synthesized in reactions of **26** (PAnthP) with Pt(L)(OTf)_2_ (L = diphosphine, OTf) (Figure 46). To understand the effect of the number of Pt ions on the extent of perturbation, a mononuclear analog **133** was also prepared. The UV–vis absorption spectra of **133** and PAnth displayed moderately intense vibronic bands at around 320–440 nm. The spectra of the binuclear complexes **130**–**132** were different from that of **133**. The spectra of the *syn*-isomers **130a**, **131a**, and **132a** displayed two intense overlapping absorption bands at 320–520 nm. The *anti*-isomers **130b** and **132b** also displayed two intense bands in a similar spectral range (300–500 nm). The emission energies in degassed DCM at room temperature followed the order **133** > **131b**, **132b** > **130a**, **131a**, and **132a** [62]. 

9-(Dihydrophosphino)anthracene **134** was prepared in two steps starting from 9-bromoanthracene **1**, which was next was converted to 9-(dihalophosphino)anthracenes **135** (X = Cl, Br) via the Grignard reagent **136**, which reacted with PCl_3_ or PBr_3_, respectively. Next, reduction of the latter with 2 equiv. of LiAlH_4_ in diethyl ether at −78 °C and then reflux for 1 h delivered **134**. In the reaction of the dilithium derivative of **134** with 1,2-dichloroethane, Kubiak and co-workers obtained 9-(1-phosphirano)anthracene **13**. Then, the reaction of **13** with 0.5 equiv. of PtCl_2_(1,5-cyclooctadiene) gave the platinum complex, *cis*-dichlorobis[1-(9-anthracene)phosphirano]platinum(II) **137** (Figure 47) [63]. The complex **137** displayed novel intramolecular *π*-stacking interactions between the anthracene ring systems. 

The same research group synthesized other platinum complexes, such as bis[1-(9-anthracene)phosphirano]dithiolateplatinum complexes **138a**–**d**, in the reaction of **137** with appropriate potassium-ethylene-2,2-dithiolates **139a**–**d** containing two electron-withdrawing groups (EWGs) in positions 1, such as methoxycarbonyl, ethoxycarbonyl and cyano groups. The final products **138a**–**d** were obtained in CH_2_Cl_2_/MeOH mixture after 18 h at room temperature in high yields. X-ray studies of the complexes displayed the intra- or intermolecular anthracene ring of the *cis*-bis{1-(9-anthracene)phosphirane} stacked structures (Figure 48) [64]. 

All of the platinum complexes **138a**–**d** reported emitted light at low temperatures in the solid state. Complexes **138a**–**d** exhibited a strong green fluorescence at 530 nm at low temperatures in the solid state. Moreover, the complex **138d** strongly emitted blue light in the THF or benzene solution at 450 nm after excitation at 420 nm. The blue emission of the complex **138d** with two cyano groups and a very small Stokes shift was similar to that observed for free 9-(1-phosphirano)anthracene and anthracene rings. 

The pincer complexes **140**, **141**, and **142** were synthesized by irradiating the cyclometalated complex **122c** in the presence of O_2_, which led to oxidations of the anthryl ring (Figure 49). The first photoproduct, a Pt(II)-9,10-endoperoxide complex **143**, was converted photochemically to the Pt(II)-9-hydroxyanthrone complex **140**, which was further oxygenated to the Pt(II)-hemiketal **141**. The oxidation of **140**, which could be accelerated by light irradiation, probably involved a Pt(II)-anthraquinone intermediate. The Pt(II)-hemiketal **141** underwent acid-catalyzed ketalization to form a binuclear Pt(II) 2-diketal **142**. The structures were characterized by NMR and single-crystal X-ray diffraction. All complexes possessed similar absorption spectra, showing a moderately intense vibronic band at 390–480 nm (λ_max_ = 454 nm, ε_max_ = 7.6–9.2 × 10^3^ M^−1^ cm^−1^) and a very intense band at 280 nm (ε_max_ = 5.2–5.9 × 10^4^ M^−1^ cm^−1^). The Pt complexes were also luminescent in solution and in the solid state at room temperature. Irradiating degassed CH_2_Cl_2_ solutions of the complexes at 390 nm resulted in an emission band at λ_max_ = 474 nm with a vibronic shoulder at 520 nm [65]. 

### 2.4. Phosphoranes (AnthPR_2_X_2_) (X = F) 

Yamaguchi et al. [26] reported the synthesis and photochemical characterization of tri(9-anthryl)difluorophosphorane **144** obtained from the reaction of xenon difluoride with tri(9-anthryl)phosphine **8**. The synthesis of **144** is presented in Figure 50. 

The authors proved that the fluorescence intensity is attributed to the coordination number of phosphorus. The tri-coordinated **8** exhibited a weak fluorescence while pentacoordinated **18** showed a significant fluorescence. 

### 2.5. Phospinimines (AnthR_2_P = N-R^1^) and Phosphiniminium Derivatives (AnthR_2_P = NH_2_^+^) 

Jurisson and co-workers synthesized the *N*-protected phosphinimine **145** and its phosphiniminium ion pairs **146a** and **146b** with [ReO^4−]^ and [TcO^4–^] anions. The phosphinimine **145** was fluorescent but the addition of [TcO_4_^−^] or [ReO_4_^−^] anions to **145** did not change the original spectrum in terms of the overall spectral features or intensity. In addition, the anthracene molecule scintillated in the presence of [^99^TcO^4−^], making it a possible reporter group for a scintillation sensor using this molecule (Figure 51) [66]. 

### 2.6. Phosphonium Salts (AnthPR_3_^+^) 

Usually, phosphonium salts are obtained from phosphines by quaternization of a tricoordinated phosphorus atom with a free electron pair. 

The D–π–A type of phosphonium salts, in which electron acceptor (A = ^+^PR_3_) and donor (D = NPh_2_) groups were linked by polarizable π-conjugated spacers, showed an intense fluorescence classically ascribed to the excited state intramolecular charge transfer (ICT). Therefore, a series of such phosphonium salts with different lengths of spacers and counterions were synthesized and characterized. The salt **147** was synthesized by the two-step approach involving the preparation of tertiary aryl phosphine from **158** followed by methylation with methyl iodide. The peak wavelengths (λ_abs_) were gradually red-shifted along with the extension of the π-spacer: π = phenylene (333 nm) < π = biphenylene (387 nm) < π = naphthylene (407 nm) < π = anthrylene (**147**, 519 nm). The extension of the π–system from phenylene to the polycyclic naphthalene and anthracene motifs in **147** caused a gradual growth of λ_em_ to 560 and 679 nm for **147** in DCM, which was, however, accompanied by a drop in the quantum yield (Figure 52) [67]. 

A metal-free synthesis of 2-anthryl phosphonium bromide **149** by the reaction of triphenylphosphine with 2-bromoanthracene **150** in refluxing phenol was developed by Huang et al. Examination of other solvents with a boiling point of around 200 °C showed that tetralin, PhCN, ethoxybenzene, or 2-chlorophenol could also produce phosphonium salts, although in lower yields (5–44%). A two-step addition-elimination mechanism was proposed, in which the second step of the bromide elimination was fast, as indicated by the deuterium experiment. The authors suggested that phenol could form a hydrogen bond with bromide, facilitating the addition of triphenylphosphine and elimination of bromide by polarizing the carbon–bromide bond, and making phenol the optimal solvent among the solvents tested (Figure 53) [68]. 

Nikitin and co-workers [69] synthesized 9-(methylphenylphosphinoyl)anthracene **151** by the oxidation of 9-(methylphenylphosphino)anthracene **152** with hydrogen peroxide in acetonitrile solution (Figure 54). The compound **151** was then converted into the corresponding phosphonium chloride **153a** and bromide **153b** using oxalyl chloride and bromide, respectively. The authors measured the exchange barriers of self- and cross-exchange of halides in phosphonium salts using the 2D EXSY NMR technique to visualize the processes. 

Tri(9-anthryl)(methyl)phosphonium iodide **154** was synthesized by Yamaguchi et al. in the quaternization reaction of the phosphine **8** with methyl iodide. The tri-coordinated **8** and tetra-coordinated derivatives **154** exhibited a weak fluorescence (Figure 55) [26]. 

Bałczewski et al. [70,71] recently presented a novel, one-pot *phospho*-Friedel–Crafts–Bradsher cyclization, which led to higher-substituted phosphonium salts **155**. In the new reaction, (*o*-diacetaloaryl)arylmethanols **156**, as the starting materials, in the presence of triphenylphosphine and acids HA, spontaneously cyclized directly to **155** under very mild reaction conditions (Figure 56). 

## 3. Synthesis and Reactions of P^III^ Acids, Their P^IV^ Tautomers, and Derivatives 

This section covers P^III^ acids derivatives containing at least one anthracene moiety linked either directly to the phosphorus atom via the P-Csp^2^ (Anth) bond (Section 3.1 and Section 3.2) or indirectly via the P-O-Csp^2^ (Anth) bond (Section 3.3). The phosphonous RP(OH)_2_, phosphinous R_2_POH, and phosphorous P(OH)_3_ free acids are the organophosphorus members of the group of substances known as the lower acids of phosphorus. These P^III^ trivalent species exist as minor tautomers in equilibrium with major P^IV^ tetravalent forms, which exhibit one less acidic function than might be expected [8]. Derivatives of P^III^ acids, such as halides, amides, and esters, may exist in stable, trivalent forms and they will be described separately in Section 3.1, Section 3.2 and Section 3.3. The P^IV^ tautomers are discussed in Section 3.4 and Section 3.5.

### 3.1. Phosphonous Acid Dihalides and Phosphinous Acid Halides (Halophosphines) (AnthPX_2_) and (Anth_2_PX) (X = F, Cl, Br) 

Halo- and dihaloanthracenes of the formula AnthPX_2_ and Anth_2_PX (X = F, Cl, Br), which contain at least one P-C bond and one or two halogen atoms, are classified as halides of the corresponding lower P^III^ acids. Syntheses of 9-(difluoro, dichloro, dibromo)anthracenes are also described in Section 2.3. 

9-Difluoroanthracene **127** was synthesized by Schmutzler et al. starting from 9-bromoanthracene **1** and chlorodifluorophosphine in a 96% yield (Figure 57) [10]. 

9-(Difluorophosphino)anthracene **127** was also employed by the group of Schmutzler in further investigations. They used 9-(dihydrophosphino)anthracene **14** and irradiated it with a mercury lamp in toluene for 22.5 h to obtain two isomeric dimers **157a** and **157b** in a 2:1 ratio, which could be observed in ^31^P NMR. Irradiation of 9-(difluorophosphino)anthracene **127** under the same conditions gave only one dimeric isomer **126**. Hydrogenation reaction of the latter with 10 equiv. of LiAlH_4_ in diethyl ether at reflux for 24 h delivered a single isomer **157a** in a 93% yield (Figure 58) [10]. 

Kirst and et al. [72] synthesized 9-(dichlorophosphino)anthracene **18** by the reaction of PCl_3_ with the organozinc compound **158**. The latter was obtained from 9-bromoanthracene **1**, which was first lithiated with *n*-butyllithium to obtain **159** and then submitted to the Li/Zn transmetallation with dry ZnCl_2_ (Figure 59). 

9-(Dichlorophosphino)anthracene **18** was further utilized by the group of Schmutzler in the preparation of cyclic (*P*-anthrylphosphino)phosphonium chloride remaining in equilibrium **160a**/**160b** in the reaction of 9-(dichlorophosphino)anthracene **18** and *N*-[*tert*-butyl(phenyl)phosphino]-*N*,*N*-dimethyl-*N*-(trimethylsilyl)urea **161** in CH_2_Cl_2_ at room temperature in a 46% yield. The existence in solution of the equilibrium between the ionic structure **160a** and the covalent form **160b** was observed. Next, the chloride **160a**/**160b** was converted into the corresponding phosphonium tetraphenylborate **162** by treatment of NaBPh_4_ in CH_2_Cl_2_/CH_3_CN in a 50% yield (Figure 60) [73]. 

Schmutzler and co-workers presented a synthesis of 9-(dichlorophosphino)anthracene **18** and 9-(anthrylchlorophosphino)anthracene **163** from 9-bromoanthracene **1** using a large excess of PCl_3_ (19 equiv.) in a 48% yield. The resulting mixture of **18** and **163** was reduced with 5.4 equiv. of LiAlH_4_ in diethyl ether at reflux to obtain 9-(dihydrophosphino)anthracene **14** and (anthrylhydrophosphino)anthracene **163**. Next, pure **14** was reacted with 1 equiv. of Os_3_(CO)_11_(CH_3_CN) in CH_2_Cl_2_ at room temperature to give triosmiumdodecacarbonyl cluster **164** quantitatively (Figure 61) [10]. 

### 3.2. Phosphonous Acid Diamides (AnthP(NR_2_)_2_) 

9-[Bis(diethylamino)phosphino]anthracene **166** and 9,10-bis[bis(diethylamino)phosphino]anthracene **167** were prepared by Tokitoh and co-workers starting from 9-bromoanthracenes **1** and **42**, which were first lithiated with *n*-butyllithium and then reacted with bis(diethylamino)chlorophosphine. The resulting anthracenes **166** and **167** were transformed to 9-(dichlorophosphino)anthracene **18** and 9,10- bis(dichlorophosphino)anthracene **168** using hydrogen chloride in diethyl ether (Figure 62) [74]. 

### 3.3. Phosphorous Acid Esters (Phosphites) (AnthOP(OR)_2_) 

This subsection covers P^III^ acids derivatives containing one anthracene moiety linked indirectly to the phosphorus atom via the P-O-Csp^2^ (Anth) bond. 

Kloß et al. conducted a study of numerous phosphite-based ligands for rhodium catalysts, which were used in hydroformylation reactions [3]. The authors revealed that anthryl phosphites were susceptible to hydrolysis, which limis their use for the synthesis of catalysts. Therefore, they synthesized relatively stable phosphites, one of which was the phosphite **169**. 

The latter, as a solid, was synthesized by a procedure involving the treatment of benzopinacol **170** with phosphorus trichloride to give chlorophosphite **171**, followed by the addition of lithium anthr-9-olate **172** (Figure 63). 

Implemented in a rhodium catalyst, it exhibited high activity towards hydroformylation. The ligand turned out to be relatively stable under hydrolysis conditions. 

### 3.4. Phosphonous Acid P^IV^ Tautomers (H-phosphinic Acids) (AnthP(O)H(OH)) 

The synthesis of the ester of the P^III^ tautomeric form of phosphonous acid, i.e., diphenyl 9-anthrylphosphonite **2**, is mentioned in Section 2. 

Schmutzler and co-workers presented the hydrolysis of 9-(dichlorophosphino)anthracene **18** in CH_2_Cl_2_ with water at room temperature, which gave anthryl-*H*-phosphinic acid **173** in a 77% yield (Figure 64) [10]. 

Yakhvarov et al. reported the synthesis of the first example of dinuclear nickel complex **20** with the bridging anthr-9-yl-P(H)O_2_ ligands. Anthr-9-yl-phosphinic acid **173** in the reaction with NiBr_2_(bpy)_2_ in dimethylformamide after 8 days at room temperature gave the nickel complex **174** in a 46% yield (Figure 65) [75]. 

The same authors reported the formation of the first example of a neutral dinuclear cobalt complex **175** formed in the reaction of cobalt dibromide with 2,2*’*-bipyridine (bpy) and 9-anthrylphosphinic acid **173** (Figure 66) [76]. 

Stawinski and co-workers reported a microwave-assisted (MW) synthesis of a series monoaryl-*H*-phosphinic acids, including anthr-9-yl-*H*-phosphinic acid **173** [77]. The microwave-assisted cross-coupling of 9-bromoanthracene **1** and anilinium *H*-phosphinate **176** was catalyzed by 3 mol% Pd_2_(dba)_3_ CHCl_3_/Xantphos^®^ as a supporting ligand and was carried out in the presence of 2.5 equiv. of triethylamine. Irradiation of the mixture with a microwave (MW) for 5 min at 120 °C produced *H*-phosphinic acid **173** in an 82% yield (Figure 67). 

Trofimov and co-workers reported another synthesis of anthr-9-yl-phosphinic acid **173** in the reaction of 9-bromoanthracene **1** with elemental phosphorus in a superbasic medium [78]. The authors treated **1** with 3.3 equiv. of phosphorus red in DMSO and the mixture of 4.8 equiv. of KOH and H_2_O as a superbase at 60 °C for 3 h (Figure 68). In this case, the yield of **173** was only 10%. 

### 3.5. Phosphinous Acid P^IV^ Tautomers (H-phosphine Oxides) (Anth_2_P(O)H) 

The synthesis of esters of the P^III^ tautomeric form of phosphinous acid, i.e., phenyl 9-anthryl(1-naphthyl)phosphinite **3** and phenyl dianthrylphosphinite **6**, is mentioned in Section 2. 

The 1-(phosphino)-1,4-diphenyl-1,3-butadiene moiety, incorporated with a dibenzobarrelene skeleton in **177**, was synthesized by Ishii and co-workers [79]. They started the synthesis from lithiation of the starting reagent **178** with *n*-butyllithium to obtain the organolithium intermediate **179** followed by treatment of the latter with 9-dichlorophosphinoanthracene **18** to obtain the key precursor (*Z*)-1-(9-anthrylchlorophosphino)butenyne **180**. Then, the dibenzobarrelene structure **181** was obtained by an intramolecular [4+2] cycloaddition reaction of **180** (Figure 69). Further hydrolysis of **181** gave the secondary phosphine oxide **177**, which exhibited a long-wavelength absorption (λ_abs_ = 355 nm) and emission (λ_em_ **=** 442 nm). 

## 4. Synthesis and Reactions of Phosphonic Acids (AnthP(O)(OH)_2_) and Phosphonates (AnthP(O)(OR)_2_) (Anth = Anthryl) 

In this section, P^IV^ organophosphorus-substituted acenes with one P-Csp^2^ (Anth) bond, two P-O, and one P=O bonds are reviewed. Hence, this section includes phosphonic acids and their esters. Interestingly, no thio-and seleno phosphonic acids AnthP(X)(YH)_2_ and the corresponding hetero-phosphonates AnthP(X)(YR)_2_ (Anth = anthryl), (X, Y = S, Se) were reported in the review period. 

The synthesis of a series of anthracenes substituted in position 2 with diethoxyphosphoryl groups was described by French and coworkers (Figure 70) [80]. The Arbuzov reaction of 2-bromo-anthracene **150** with triethylphosphite, catalyzed by nickel bromide, proceeded in refluxing mesitylene for 20 h and led to 2-(diethoxyphosphoryl)anthracene **182**. Then, the latter was transformed into the disilyl diester by treatment with bromotrimethylsilane in dichloromethane and next hydrolyzed to the free acid **183** with methanol. Dissolving **183** in an aqueous solution of a stoichiometric amount of sodium hydroxide gave the sodium salt **184** quantitatively. 

Then, the authors investigated the spectroscopic properties of the obtained compounds, including the absorbance, fluorescence, and quantum yields Φ [80]. A slight blue shift was observed after the conversion of phosphonate ester into phosphonic acid and also when the phosphonic acid was converted to the corresponding sodium salt. Shifts of 53, 53, and 49 nm were observed for **182**, **183**, and **184**, respectively. Compounds **182**, **183**, and **184** showed quantum yields of fluorescence Φ = 33%, 40%, and 0%, respectively. Additionally, compounds **182**, **183**, and **184** formed micelles in water. 

Nagode and co-workers [81] synthesized 1-hydroxy-4-phenyl-2-(dimethoxyphosphoryl)anthracene **185** using α-diazophosphonate **186**, phenylacetylene **187**, Hantzsch ester, and tetrabutylammonium bromide (TBAB) in dichloroethane under blue LED irradiation. This reaction was conducted at room temperature for 6–8 h and the product **185** was obtained in a 70% yield (Figure 71). 

Shu et al. [82] described a new synthetic method for the preparation of 2-hydroxy-1-(dimethoxyphosphoryl)anthracene **188** in the reaction of an imine derivative of azomethine **189** and dimethyl diazophosphonate **190** in the presence of inorganic additives (Figure 72). The reaction was carried out at 100 °C and the compound **188** was obtained in a 29% yield. 

Nakamura and co-workers [83] studied the photolysis reactions of di(anthr-9-yl)metylphosphonate **191**. The authors demonstrated that upon irradiation with monochromatic light at 365 nm, **191** underwent cyclization to **192** (Figure 73), similarly to the compound **230** (see below Scheme 87). 

Bessmertnykh and co-authors presented a direct synthesis of 9-(diethoxyphosphoryl)anthracene **193** bearing an amino group on the aromatic ring at the position 2 in a Hirao reaction. The authors carried out a reaction of 2-amino-9-bromoanthracene **194** with diethyl phosphite in the presence of *N*,*N*-dicyclohexylmethylamine in refluxing ethanol for 48 h and using catalytic amounts of palladium acetate (5 mol %) and triphenylphosphine (15 mol %). The outcome of this reaction depended on the stoichiometry used (Figure 74) [84]. 

Leenstra and co-workers synthesized zirconium bis-(2-anthrylphosphonate) [Zr(Anth)_2_] **196** by mixing zirconyl chloride with hydrofluoric acid, sodium hydroxide, and 2-naphthylphosphonic acid **183** in water for 5 days at reflux (Figure 75) [85]. 

The same authors showed the readily excimer formation of zirconium bis-(2-anthrylphosphonate) [Zr(Anth)_2_] **196** as a powdered solid in glycerol and its precursor, 2-anthrylphosphonic acid **183**, in methanol solution (Figure 75) [86]. The fluorescence spectrum of **196** [Zr(Anth)_2_] had a broad emission band with a maximum at 448 nm in the solid. Additionally, the authors observed that [Zr(Anth)_2_] **196** did not display a time-dependent quenching of the fluorescence emission, suggesting that the photodimerization reaction of the anthracene group to bianthryl did not exist in the solid state. 

Zhou and coworkers [6] investigated five anthracene-based bis(phosphonic acids), of which three **197a**, **197b**, and **197c** possessed the direct P-C_sp_^2^ bonding. They were prepared according to the literature procedures cited there (Figure 76). Thus, 4,4-(anthracene-9,10-diyl)bis(4,1-phenylene) bis(phosphonic acid) **197a** was synthesized by treatment of 1,4-dibromobenzene **198** with *n*-butyllithium, followed by the addition of the resulting 4-bromo-1-lithiobenzene to anthraquinone and subsequent reduction of the resulting anthryl dialcohol to give **199**. Then, the Arbuzov-type reaction followed by hydrolysis with trimethylsilyl bromide (TMS-Br) of the obtained bis(phosphonate) **200** gave the bis(phosphonic acid) **197a** in a 36 % yield (Figure 76) [87]. Using this procedure and 1,3-dibromobenzene as the starting material, the authors obtained the corresponding regioisomeric 4,4′-(anthracene-9,10-diyl)bis(3,1-phenylene)bis(phosphonic acid) **197b** [88]. Anthracene-9,10-bis(phosphonic acid) **197c** was prepared analogously, based on the procedure by Pramanik et al. of the synthesis of the corresponding tetraethyl ester **201** (Figure 77) [7], which was next hydrolyzed in this work with TMS-Br to give **197c**. Compounds **197a** and **197c** exhibited red-shift fluorescence. They were deposited on a zirconium dioxide layer as triplet-triplet annihilation acceptors. 

M. Pramanik et al. [7] synthesized **201** from 9,10-dibromoanthracene **42** in a 37% yield, at high temperature, using an excess of triethyl phosphite and NiBr_2_ in 1,3-diisopropylbenzene (Figure 77). The compound **201**, in the form of fluorescent organic nanoparticles, was explored as a selective anticancer candidate by apoptosis-mediated cancer therapy towards U937 cells. 

In another study, Yazji et al. [89] obtained 9,10-diphenyl-2,6-bis(phosphonic acids) **202a** and **202b** (Figure 78) from 9,10-diaryl-2,6-dibromoanthracenes **203a** and **203b** which next were transformed to the corresponding 2,6-dilitio derivatives with *t*-butyllithium, and then reacted with diethyl phosphorochloridate to give bis(phosphonates) **204a** and **204b**, which were finally converted, by the hydrolysis of the diester, to the corresponding bis(phosphonic acids) **202a** and **202b** (Figure 78). 

The authors investigated the use of these compounds in thin films deposited on a silicon dioxide surface, acting as nucleation sites for pentacene crystallization. This study suggested that high optical anisotropy of pentacene, crystallized on such films, indicated that compounds **202a** and **202b** changed the silicon dioxide surface into a lattice, which induced the nucleation of pentacene. A similar study involving **202a** and **202b** was also conducted by Cattani-Scholz and co-workers [90], who synthesized these compounds in the same manner as the group of Yazji et al. 

Kabachnik and coworkers demonstrated a synthesis of tetramethyl bis(phosphonate) **205** starting from 9,10-dibromo-anthracene **42** and dimethyl phosphite (Figure 79) [91]. This reaction was catalyzed by palladium acetate/triphenylphosphine and was carried out under bi-phasic conditions for 30 h at 70–80 °C in acetonitrile in the presence of K_2_CO_3_ as a base and benzyltriethylammonium chloride (BTEAC) as a phase-transfer catalyst (PTC) to give the desired product **205** in a 70% yield. 

Next, the authors transformed both phosphonate ester groups in **205** to the corresponding bis(phosphonic acid) **206** by refluxing it in an aqueous solution of hydrochloric acid for 2 h in an 80% yield (Figure 79) [91]. 

Organic thin-film transistors based on pentacene as a semiconductor were fabricated on silicon. A self-assembled monolayer derived from the phosphonate (SAMP) **207a** showed an improvement over monolayers using octadecylsilane and other phosphonates (Figure 5). These devices had substantially reduced trap states, on/off ratios of 10^8^, subthreshold slopes of 0.2 V/decade, and substantially uniform threshold voltages of −4.5 V across a large number of devices [92]. 

Good device characteristics were also measured for the monolayer **207b**, in which the calculated molecular spacings were about 0.7 nm. This created channels that were on the order of the “thickness” of an aromatic π system, and which could allow intercalation of pentacene units, favoring a π-stacking motif for this first pentacene layer [93]. 

Tornow and co-workers synthesized SAMPs **208c**–**e** and self-assembled organophosphonate duplexes **209c**,**d** ensemble on nanometer-thick SiO_2_-coated, highly doped silicon electrodes (Figure 5) [5]. 

Most of the reviewed papers in the previous sections discussed organophosphorus-substituted anthracenes that did not contain other substituents or anthracenes with a very low degree of substitution. This problem also concerns a group of phosphonates and phosphonic acids. Bałczewski et al. [70,71] recently presented a new *phospho*-Friedel–Crafts–Bradsher cyclization, which enabled the synthesis of highly substituted anthracenes **210**. In this new reaction, (*o*-diacetaloaryl)arylmethylphosphonates **211** were cyclized under very mild conditions at room temperature to give 10-(dialkoxyphosphorylanthracenes **212** in a 70–98% yield. The latter were hydrolyzed to the corresponding phosphonic acids **210** in a 67–85% yield (Figure 80). 

## 5. Synthesis and Reactions of Phosphoric Acids and Phosphates (AnthOP(=O)(OR)_2_) (R = H, alkyl, aryl) 

Unlike previous sections (except Section 3.3), which reviewed the synthesis and reactions of compounds containing a phosphorus atom linked to the anthracene moiety directly by the P-Csp^2^ (Anth) bond, this section and Section 3.3 cover anthracenes that are bonded to the phosphorus atom indirectly via an oxygen atom by the P -O-Csp^2^ (Anth) bond. Interestingly, no hetero-analogs of phosphates were reported in the reviewed period. 

Buckland and Davidson [94] investigated the photo-oxidation of 10-diethoxyphosphoryloxyanthracene **213**, which represents a group of acenyl phosphates. The authors showed that in the presence of oxygen, the photochemically labile phosphate **213**, dissolved in acetonitrile, could be oxidized to anthraquinone **50** and diethyl hydrogen phosphate **214** upon irradiation with a 365 nm monochromatic light (Figure 81). 

Yamashita et al. [95] conducted a study in which 1,8-dimethoxyanthr-9-ol **215** was used for the synthesis of 9-bromo-1,8-dimethoxyanthracene **216**. The former was treated with diethyl phosphorochloridate in the presence of sodium hydride, yielding 9-diethoxyphosphoryloxy-1,8-dimethoxy-anthracene **217**. Then, the phosphate/lithium exchange with Li_(DTBB)_ (DTBB = 4,4′-di-tert-butylbiphenyl) followed by bromination with 1,2-dibromo-1,1,2,2-tetrafluoroethane gave **216** as a pale-yellow solid in a 30 % yield (Figure 82). 

Another study, carried out by the group of Yamashita and co-workers [96], showed further application of the Li_(DTBB)_ system towards the substrate **217**. When 2.5 equiv. of Li_(DTBB)_ was used, the reaction followed the path from Figure 82. However, when an excess (10 equiv.) of the Li_(DTBB)_ reagent was used, the phosphate **217** was transformed to 1,8,9-trilithioanthracene **218**, which then underwent bromination with 1,2-dibromo-1,1,2,2-tetrafluoroethane to give 1,8,9-tribromoanthracene **219** (Figure 83). 

Takizawa et. al. [4] discovered an enantioselective, oxidative-coupling reaction of anthr-2-ol **220** with the chiral vanadium complexes **221a** and **221b** to deliver bianthrol **222**, which was then esterified with phosphoric acid to give 4,4’-bianthryl phosphoric acid **223** (Figure 84). 

The authors investigated the catalytic properties of **223** in the Diels–Alder reaction of 2-cyclohexenone with aldimines; however, it gave only racemic mixtures. 

Another application of anthryl phosphates in the Diels–Alder reaction was performed by Meek and Koh [97], who described the synthesis of 10-bromo-9-(dimethoxyphosphoryloxy)anthracene **224** obtained from 10,10-dibromoanthrone **225** and trimethyl phosphite. Next, they applied **224** as a diene in the Diels–Alder reaction with acrylic acid and maleic anhydride, which delivered adducts **226** and **227**, respectively (Figure 85). 

Then, the authors [97] described the reaction of anthraquinone anil **228** with trimethyl phosphite, yielding 9-dimethoxyphosphoryloxy-10-(phenylamino)anthracene **229** as yellow needles (Figure 86).

Nakamura and co-workers [83] studied the photolysis reactions of tri(anthr-9-yl)phosphate **230**. The authors demonstrated that upon irradiation with a 365 nm monochromatic light, **230** underwent cyclization to **231** (Figure 87). Interestingly, the product **231** could be transformed back to **230** upon irradiation with a 254 nm monochromatic light. 

## 6. Synthesis and Reactions of Diphosphenes (Anth(P=PR)) and Derivatives 

In addition to anthracenes substituted by one, two, or three organophosphorus groups with one phosphorus atom in each group, which have been described in previous sections, this section reviews anthracenes substituted by organophosphorus groups containing two or three phosphorus atoms. 

9-(Diphospheno)anthracenes **232** and 9,10-bis(diphospheno)anthracenes **233**, presumably the first stable (diphospheno)anthracenes, were synthesized by Tokitoh and co-workers [74,98]. The 2,4,6- tris[bis(trimethylsilyl)methyl]phenyl (Tbt) and 2,6-bis[bis(trimethylsilyl)methyl]-4-[tris(trimethylsilyl)methyl]phenyl (Bbt) groups, reported by Yoshifuji and co-workers, were employed for stabilization of these molecules [99]. First, 9-dichlorophosphinoanthracene **18** and 9,10-bis(dichlorophosphino)anthracene **234** were readily prepared in moderate yields from 9-bromoanthracenes **1** and **42**. Next, the condensation reaction of TbtPH_2_ **235a** and BbtPH_2_ **235b** with 9-dichlorophosphinoanthracene **18** in the presence of DBU (DBU = 1,8-diazabicyclo[5.4.0]undec-7-ene) as a base afforded 9-diphosphenoanthracenes **232a** and **232b** as stable red crystals in 71 and 77% yields, respectively (Figure 88). 9,10-Bis(diphospheno)anthracenes **233a** and **233b** were synthesized in a ca. 20% yield in a manner similar to the synthesis of 9-diphosphenoanthracenes **232a** and **232b** using 9,10-bis(dichlorophosphino)anthracene **234** instead of 9-(dichlorophosphino)anthracene **18**. The UV-vis spectra of **232** and **233** revealed the electronic communication between the anthryl and P=P units, which was also supported by the TD-DFT calculations. The monodiphosphene derivative **232a** exhibited a weak fluorescence in hexane solution, whereas the bis(diphosphene) derivatives **233** displayed no appreciable luminescence under the same conditions. The compounds **232a**,**b** and **233a**,**b** showed absorption maxima at 380–400, 380–400, 403–426, and 404–427 nm, accordingly. 

The same research group also examined the specific reactions of the 9-(diphospheno)anthracene **232a**, including sulfonation, selenation, and attempted telluration with tributylphosphine telluride (Figure 89) [74,100]. 

Thus, sulfonation of **232a** gave thiadiphosphirane **236** while selenation delivered selenadiphosphirane **237** in good yields. 

Surprisingly, treatment of **232a** with tributylphosphine telluride did not result in telluration via the tellurium transfer. Instead, the reaction yielded triphosphirane **238** as yellow crystals and the diphosphene derivative **239** as red crystals. 

## 7. Conclusions

In this review, covering the period 1968–2022, the synthetic methods, reactions, and applications of acenes were discussed. This review revealed that phosphorus-substituted acenes with a number of benzene rings greater than three remain unknown. This opens the way for the development of new syntheses of longer acenes and insights into the properties and novel applications of such materials. Based on the current knowledge of the properties of multi-ring fused aromatics, it can be predicted that such acenes, especially electron-tunable (P^III^, P^IV^, P^V^) phosphorus-substituted tetracenes and pentacenes, will find more effective applications than lower analogs, mainly in optoelectronics. In particular, solid-anchored phosphonic acids that form monolayers may provide an example of such an application [5,92,93]. 

The second characteristic of the reviewed compounds was the low degree of substitution of aromatic rings by other substituents than organophosphorus groups and, in particular, most of anthracene moieties were unsubstituted. Apart from our preliminary work [70,71], this review showed, practically, a lack of works devoted to the highly substituted acene systems. It is noteworthy that highly substituted acenes containing thioorganic substituents showed extremely high thermal and photochemical stabilities, properties that would be interesting to verify for acenes with organophosphorus substituents [101,102]. Furthermore, the absence of hetero(S, Se)-analogs of phosphonates and phosphates was recorded during the reviewed period, which additionally opens the way for further research in this area.

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
