# Peer review of "Comprehensive Review on Synthesis, Properties, and Applications of Phosphorus (PIII, PIV, PV) Substituted Acenes with More Than Two Fused Benzene Rings"

_molecules, 2022, doi:10.3390/molecules27196611_

Round 1

Reviewer 1 Report

This is a very nice review on linearly-fused aromatics with P-function by Prof. Balcewski et al covering the last 50 years. Their own results were also incorporated. The ms will be acceptable after the following minor revision:

         Page 2/line 78: “phosphines AnthPR2” is awkward (in the subtitle). Please use parentheses: “phosphines (AnthPR2)” or “phosphines of type AnthPR2”.

         Scheme 1 and throughout: please specify “rt” between 18-32 °C.

         Page 3/line 108: insert “,” before “which”.

         Scheme 2: Why are there two steps in the 7 ® 10 conversion?

         Can the intermediate of the 11 ® 12 conversion be given in Scheme 3?

         Why was it necessary to replace Cl to F in Scheme 7?

         Page 9: Subtitle: “2.1 Phosphine oxides AnthP(=O)R2…”is again awkward.

         On page 3/line 109: “Ph2PCl” was written, while on page 10/line 265: “chlorodiphenylphosphine” was can be seen. Please unify throughout!

         Scheme 17: structures 8 and 45 are awkward due to the misleading valency of P.

         Please name the transformation in Scheme 22 as a Hirao reaction.

         Page 15/line 421: instead of “obtained compounds” write “compounds obtained”.

         “2.2 Phosphine boranes AnthPR2•BH3”: modify.

         Scheme 28: please provide conditions

         “2.3 Phosphine-metal complexes AnthPR2-met”; “met” is also awkward.

         Page 17/line 478: the dot in the complex should be bigger.

         What causes the stereoselectivity in compound 90?

         Scheme 33: temperature?

         Scheme 44: temperature?

         Scheme 45: temperature?

         Scheme 51: temperature?

         “2.6 Phosphonium salts AnthPR3+” to be refined.

         Scheme 52: temperature?

         Scheme 57: temperature?

         Scheme 59: temperature?

         Scheme 62: temperature?

         Subtitle 3.3: to be refined.

         Subtitle 3.4: to be refined.

         Subtitle 3.5: to be refined.

         Subtitle 4: to be refined.

         Scheme 74 also involves a Hirao reaction.

         Page 41/line 1042: insert “,” after t-BuLi.

         Line 1043: instead of “diester hydrolysis” write “hydrolysis of the diester”.

         Page 41/line 1052: delete “,” after “A similar study”

line 1053: insert ”,” before “who”.

line 1059: BTEAC” is rather “TEBAC” (also in Scheme 79)

         Subtitle 5 to be refined.

         Page 44/line 1114 and Schemes 82 and 83 “Li(DTBB)” is awkward.

         Scheme 84: give conditions for the ring closure by H3PO4.

         Page 45/ line 1138: insert “,” before who.

         Scheme 85: the 225 ® 224 conversion would deserve a comment. Also, please include conditions.

         Scheme 86: Also, please include conditions.

         Subtitle 6: to be refined.

Finally, it is stressed that this is a very interesting and useful review embracing organoP chemistry and Li-organic chemistry.

Author Response

Response to Reviewer 1 Comments

This is a very nice review on linearly-fused aromatics with P-function by Prof. Bałczewski et al covering the last 50 years. Their own results were also incorporated. The ms will be acceptable after the following minor revision:

Point 1: Page 2/line 78: “phosphines AnthPR2” is awkward (in the subtitle). Please use parentheses: “phosphines (AnthPR2)” or “phosphines of type AnthPR2”.

Response 1: To unify, we used parentheses, like e.g: “phosphines (AnthPR2)” here and in other titles and subtitles of this manuscript indicated by the Reviewer 1, and even in cases not indicated by the Reviewer 1.

Point 2: Scheme 1 and throughout: please specify “rt” between 18-32 °C.

Response 2: The authors used in the original papers “rt” term.

Point 3: Page 3/line 108: insert “,” before “which”.

Response 3: We added “,” before “which”.

Point 4: Scheme 2: Why are there two steps in the 7 to 10 conversion?

Response 4: The authors added the reagent in two portions.

Point 5: Can the intermediate of the 11 to 12 conversion be given in Scheme 3?

Response 5: We added the intermediate in Scheme 3.

Point 6: Why was it necessary to replace Cl to F in Scheme 7?

Response 6: The synthesis from dichloro derivative was unsuccessful.

Point 7: Page 9: Subtitle: “2.1 Phosphine oxides AnthP(=O)R2…”is again awkward.

Response 7: We added parentheses in the title: “Phosphine oxides (AnthP(=O)R2), phosphine sulfides (AnthP(=S)R2), phosphine selenides (AnthP(=Se)R2)”.

Point 8: On page 3/line 109: “Ph2PCl” was written, while on page 10/line 265: “chlorodiphenylphosphine” was can be seen. Please unify throughout!

Response 8: We unified the name into “chlorodiphenylphosphine”.

Point 9: Scheme 17: structures 8 and 45 are awkward due to the misleading valency of P.

Response 9: The scheme has been corrected.

Point 10: Please name the transformation in Scheme 22 as a Hirao reaction.

Response 10: We added the name reaction: “Hirao”.

Point 11: Page 15/line 421: instead of “obtained compounds” write “compounds obtained”.

Response 11: We changed the phrase to “compounds obtained”.

Point 12: “2.2 Phosphine boranes AnthPR2•BH3”: modify.

Response 12: We changed the title and added ”•”. Moreover, we added parentheses in (AnthPR2•BH3).

Point 13: Scheme 28: please provide conditions.

Response 13: We added the reactin conditions.

Point 14: “2.3 Phosphine-metal complexes AnthPR2-met”; “met” is also awkward.

Response 14: We changed “Met” to “Metal”.

Point 15: Page 17/line 478: the dot in the complex should be bigger.

Response 15: We increased the size of the dot in the complex.

Point 16: What causes the stereoselectivity in compound 90?

Response 16: The compund 90 is achiral. The scheme was corrected.

Point 17: Scheme 33: temperature?

Response 17: We added the reaction temperature.

Point 18: Scheme 44: temperature?

Response 18: We added temperatures in the Scheme.

Point 19: Scheme 45: temperature?

Response 19: We added the temperature in the Scheme.

Point 20: Scheme 51: temperature?

Response 20: We added the temperatures in the Scheme.

Point 21: “2.6 Phosphonium salts AnthPR3+” to be refined.

Response 21: We added parentheses in the title.

Point 22: Scheme 52: temperature?

Response 22: We added the temperature in the Scheme.

Point 23: Scheme 57: temperature?

Response 23: We added the temperature in the Scheme.

Point 24: Scheme 59: temperature?

Response 24: We added temperatures in the Scheme.

Point 25: Scheme 62: temperature?

Response 25: We added the temperature in the Scheme.

Point 26: Subtitle 3.3: to be refined.

Response 26: We added parentheses in the subtitle.

Point 27: Subtitle 3.4: to be refined.

Response 27: We added parentheses in the subtitle.

Point 28: Subtitle 3.5: to be refined.

Response 28: We added parentheses in the subtitle.

Point 29: Subtitle 4: to be refined.

Response 29: We added parentheses in the subtitle.

Point 30: Scheme 74 also involves a Hirao reaction.

Response 30: We added phrase “in the Hirao reaction”.

Point 31: Page 41/line 1042: insert “,” after t-BuLi.

Response 31: We added “,”after t-butyllithium.

Point 32: Line 1043: instead of “diester hydrolysis” write “hydrolysis of the diester”.

Response 32: We changed to “hydrolysis of the diester”.

Point 33: Page 41/line 1052: delete “,” after “A similar study”.

Response 33: We deleted “,” after “A similar study”.

Point 34: line 1053: insert ”,” before “who”.

Response 34: We added “,”before “who”.

Point 35: line 1059: BTEAC” is rather “TEBAC” (also in Scheme 79).

Response 35: BTEAC abbreviation is commonly used in the literature.

Point 36: Subtitle 5 to be refined.

Response 36: We added parentheses in the title.

Point 37: Page 44/line 1114 and Schemes 82 and 83 “Li(DTBB)” is awkward.

Response 37: “Li(DTBB)” is an anion of 4,4′-di-tert-butylbiphenyl. This abbreviation was used in literature.

Point 38: Scheme 84: give conditions for the ring closure by H3PO4.

Response 38: We added the reaction conditions. We changed number 221 to 222.

Point 39: Page 45/ line 1138: insert “,” before who.

Response 39: We added “,”before who.

Point 40: Scheme 85: the 225 to 224 conversion would deserve a comment. Also, please include conditions.

Response 40: The authors consider various mechanisms for this transformation. We added the reaction conditions.

Point 41: Scheme 86: Also, please include conditions.

Response 41: We added the reaction conditions.

Point 42: Subtitle 6: to be refined.

Response 42: We added parentheses in the subtitle.

Finally, it is stressed that this is a very interesting and useful review embracing organoP chemistry and Li-organic chemistry. 

Reviewer 2 Report

The presented Review deals with chemistry of phosphorus substituted acenes bearing more than two fused benzene rings. Such compounds have potential for their application as optoelectronic materials, therefore the comprehensive review covering synthesis and chemical modification of these compounds is highly appealing. The Review is well organized and contains balanced scope of references. I find it suitable for publication in Molecules in present form.

Author Response

The presented Review deals with chemistry of phosphorus substituted acenes bearing more than two fused benzene rings. Such compounds have potential for their application as optoelectronic materials, therefore the comprehensive review covering synthesis and chemical modification of these compounds is highly appealing. The Review is well organized and contains balanced scope of references. I find it suitable for publication in Molecules in present form.

Response 1: We would like to thank to the Reviewer 2 for his very positive and encouraging review.

Reviewer 3 Report

Title: Comprehensive review on synthesis, properties and applica-2 tions of phosphorus (PIII, PIV, PV) substituted acenes with more 3 than two fused benzene rings

The review written by Koprowski et al summarized the synthesis, reactions and applications of linearly fused aromatics (acenes) reported from over last five decades. Their characteristic feature is substitution of the aromatic system by one, two or three organophosphorus groups, which determine their properties and applications. First, the authors have given concise introduction about synthesis and reactions of phosphines AnthPR2 (Anth = anthryl) and derivatives.  The authors have divided this review into five main sections covering phosphines, PIII acid derivatives, diphosphenes, phosphonates and phosphonic acids, phosphates, and also hetero analogs of the mentioned compounds. In the subsections, there have been placed groups of compounds that can be obtained directly from the precursor included in the main section, e.g. phosphine oxides, phosphonium salts, and phosphoranes can be obtained from phosphines by oxidation, alkylation, halogenation, respectively and therefore they have been placed in the section devoted to synthesis and reactions of phosphines and derivatives. Finally, the authors discussed about synthesis and reactions of diphosphenes Anth(P=PR) and derivatives. I suggest the authors to maintain the same size for all the schemes throughout the manuscript and the authors should improve the language. Overall, the review is well written, and all the references are in proper format. I suggest publishing this review in “Molecules”.

Author Response

The review written by Koprowski et al summarized the synthesis, reactions and applications of linearly fused aromatics (acenes) reported from over last five decades. Their characteristic feature is substitution of the aromatic system by one, two or three organophosphorus groups, which determine their properties and applications. First, the authors have given concise introduction about synthesis and reactions of phosphines AnthPR2 (Anth = anthryl) and derivatives.  The authors have divided this review into five main sections covering phosphines, PIII acid derivatives, diphosphenes, phosphonates and phosphonic acids, phosphates, and also hetero analogs of the mentioned compounds. In the subsections, there have been placed groups of compounds that can be obtained directly from the precursor included in the main section, e.g. phosphine oxides, phosphonium salts, and phosphoranes can be obtained from phosphines by oxidation, alkylation, halogenation, respectively and therefore they have been placed in the section devoted to synthesis and reactions of phosphines and derivatives. Finally, the authors discussed about synthesis and reactions of diphosphenes Anth(P=PR) and derivatives. I suggest the authors to maintain the same size for all the schemes throughout the manuscript and the authors should improve the language. Overall, the review is well written, and all the references are in proper format. I suggest publishing this review in “Molecules”:

Response 1: The Reviewer 3 recommended: "editing of English language and style required" but did not specify details. His opinion is very general and finally positive.